# Dynamic Mixture Embeddings for Contextual Meta-Reinforcement Learning

## Abstract

Contextual meta-reinforcement learning (meta-RL) relies on latent task embeddings to enable rapid adaptation when faced with an unknown task. However, most methodologies rely on unimodal priors, which lack the adaptive capacity to represent complex multimodal task structure, limiting performance when faced with non-parametric variation. We introduce *Dynamic Mixture Embeddings (DME)*, a belief-based contextual meta-RL method that learns a hierarchical Gaussian-mixture Variational Autoencoder, in which mixture component parameters are conditioned on a high-level macro latent. This yields an adaptive mixture prior whose means and variances shift as more context is gathered, while training is further augmented with virtual tasks drawn from the adaptive prior. DME achieves state-of-the-art performance across the entire MetaWorld benchmark suite, designed to test adaptation under non-parametric variation.

## 1 Introduction

Contextual meta-reinforcement learning (meta-RL) fundamentally seeks to endow an agent with the ability to *infer the current task dynamics* after only a brief period of interaction, while retaining strong performance across a broader task distribution. Functionally, this enables contextual meta-RL policies to quickly adapt and re-optimise their behaviour when faced with new, unseen tasks. Recent advancements in this field have produced a mature, practical framework that is seeing use in a growing range of real-world systems, from robust autonomous driving in diverse environments (Jiang et al., 2024; Hu et al., 2025) to situational behaviour in robotics applications (Ballou et al., 2023; Shokry et al., 2024). Put together, these examples underscore meta-RL's promise as a robust, data-efficient alternative to task-specific reinforcement learning in scenarios with changing task dynamics.

The broad range of variations in environments and tasks a contextual meta-RL agent must face can be separated, using the terminology of Yu et al. (2020), into two complementary groups. In *parametric variation*, tasks differ only through continuous parameters such as limb mass, joint friction, reward weights, or observation noise. While these variations still represent a broader distribution of MDPs, contemporary algorithms are frequently able attain near-expert performance after the handful of interactions required to identify the task (Zintgraf et al., 2021b; Zhang et al., 2021). In *non-parametric variation*, the structure and semantics of the task itself alters: object sets change, new goal interactions appear, and the transition dynamics can change entirely. Although recent methods have taken strides towards addressing these challenges through approaches such as task clustering (Chu et al., 2024) and task simulation (Lee et al., 2023), empirical performance on key benchmarks remains well below those achieved in the parametric setting.

As contextual meta-RL agents begin to tackle increasingly challenging problems, the representational capacity of latent embeddings that supply the agent with task context remains a bottleneck. In complex environments, the complexity and variety of tasks can overwhelm an encoder's ability to separate qualitatively different behaviours and task regimes, leading to entangled representations that blur the boundaries between objectives. At the same time, the encoder must remain flexible enough to account for previously unseen dynamics, while still maintaining the sample efficient adaptation that underpins meta-RL as a whole. Addressing these requirements simultaneously is a core challenge that motivates the methodology presented in this work.

A popular approach towards addressing some of these concerns involves multimodal representations with Gaussian mixture embeddings (Wen et al., 2024; Lee et al., 2023). By carving latent space into distinct regions, such embeddings are able to isolate incompatible modes of behaviour and therefore provide more representational capacity than standard Gaussian embeddings. However, existing methods suffer from rigidity in mixture parameters: the number and location of components are fixed *a priori*, so when training later encounters unexpected challenges, the model must choose an existing cluster, eroding representation quality and slowing adaptation.

We address this inflexibility with **Dynamic Mixture Embeddings** (DME), a contextual meta-RL method that utilises a hierarchical Gaussian-mixture variational auto-encoder (GMVAE) whose component parameters are themselves conditioned on a macro latent that is updated online from each context window. As new evidence arrives, mixture means and covariances migrate or expand, reallocating capacity while preserving previously learned structure. The primary contributions of our work are as follows:

1. We develop a task encoder based on a hierarchical Gaussian-Mixture VAE whose component parameters are conditioned on a macro latent variable. This dynamic mixture-representation not only achieves greater expressive capacity, but also integrates well with virtual training modules (Lee & Chung, 2021), strengthening robustness to non-parametric task variation.

2. The subsequent method, DME, achieves state of the art performance on the entire Meta-World benchmark suite (Yu et al., 2020), ranging from parametric variation in individual ML1 tasks, to the challenging non-parametric ML10 and ML45 benchmarks

## 2 PROBLEM STATEMENT AND NOTATION

In this section, we state the contextual meta-RL problem and align on notation for the rest of the paper. Each RL task can be considered as an MDP $M_i = \{S, A, P^i, R^i$ drawn from a distribution $p(M)$. This distribution can contain both non-parametric and parametric task variation, but tasks from the same distribution are considered to be semantically similar and share some implicit structure across $P^i$ and $R^i$. During adaptation, at timestep $t$ the agent observes a short context $\mathbf{c}_{0:t} = \{(s_\ell, a_\ell, r_\ell, s_{\ell+1})\}_{\ell=0}^{t}$, and produces an episode-specific policy $\pi_\theta(a_t|s_t, \mathbf{c}_{0:t})$. For notational simplicity, we will often drop the subscript $t$ unless it is relevant (e.g. $\pi_\theta(a|s, \mathbf{c})$).

The meta-objective is to maximise expected return over the task distribution:

$$\max_\theta \ \mathbb{E}_{M_i \sim p(M)}\Big[\mathbb{E}_{\tau \sim \pi_{\theta_i}}\big[r_i(\tau)\big]\Big].$$

Finally, in this work we discuss a latent model with decomposition $z = (\tilde{w}, \tilde{y}, \tilde{z})$. Throughout this work, we will use $z$ to represent latent representations as a conceptual whole, while $\tilde{z}$ refers to the decomposed low-level task embedding.

## 3 RELATED WORK

**Contextual meta-RL.** Contextual meta-RL methods embed prior context into a probabilistic task representation, which in turn informs the RL policy, enabling rapid adaptation to new tasks. Traditionally, contextual meta-RL methods have employed a single unimodal latent variable as the task representation (Zhao et al., 2021; Liu et al., 2021). Adaptation is typically achieved either by posterior sampling after a few exploratory episodes (Rakelly et al., 2019; Zhang et al., 2021; Wang et al., 2024), or by a belief-based approach in which task uncertainty is explicitly conveyed to the policy (Zintgraf et al., 2021a;b; Imagawa et al., 2022). While effective, such unimodal formulations can struggle to capture complex or multi-modal task distributions. Our approach, DME, takes a belief-based approach to adaptation but introduces a hierarchical mixture prior over latent variables, thereby providing a richer representational structure.

**Mixture latent variables and virtual training in meta-RL.** Gaussian mixture latent task representations have seen increasing use across all contextual meta-RL paradigms due to their ability

to model complex, multimodal task distributions. These mixture priors have been used to explicitly partition tasks for separate embedding modules (Chu et al., 2024), improve robustness in non-stationary environments (Poiani et al., 2021; Bing et al., 2023; Wang et al., 2023), and even extend the prior non-parametrically so the number of mixture components can grow (Bing et al., 2024). Compared to previous methods, DME conditions mixture component parameters on a high-level macro latent inferred from context, allowing means and variances to move during adaptation, as the inferred value of the macro latent is updated. Although hierarchical Gaussian mixtures have been considered in the broader meta-learning literature (Zhang et al., 2023), to our knowledge DME is the first to have truly adapted this dynamic latent structure to meta-RL.

Virtual training augments these representations by simulating additional experience from learned task reconstructions. Within, task latents are sampled from the prior, before utilising a learned reward decoder to create simulated rewards, before attaching them to existing transitions to generate new context for training. This has been shown to be an effective mechanism to improve generalisation beyond the empirical training task distribution across both on-policy (Lee & Chung, 2021; Lee et al., 2023; Kim et al., 2025).and off-policy (Ajay et al., 2022; Wen et al., 2024) variants. DME follows the established template of sampling task latents and decoding rewards for stored transitions, but its synthetic tasks are drawn from adaptive mixture components, reducing dependence on tuning the number of mixtures $K$.

**Hierarchical meta-RL.** Hierarchical control has long been used to tackle long horizons and hard exploration by decomposing behaviour across levels: a high-level controller proposes sub-behaviours or options and a low-level policy executes them (Bacon et al., 2017; Vezhnevets et al., 2017; Levy et al., 2017; Nachum et al., 2018). This abstraction seeks to concentrate exploration and credit assignment where it is most effective, utilising the slower decision-making time scales of the high-level controller to maintain consistent behaviour.

This policy-level hierarchy has also been adapted to the meta-RL problem through a variety of means. One approach abstracts sequences of actions into a stacks that are selected by a higher-level decision layer, seeking to standardise high-level choices across tasks and reduce the searchable task space during adaptation (Cho & Sun, 2024). Alternatively, instead of utilising action chains, additional information can be provided to the hierarchical meta-RL agent, such as an action-conditional skill representation (He et al., 2024), or an auxiliary inner-loop value function (Bhatia et al., 2023). In both cases, the hierarchical structure has been shown to improve sample efficiency and generalisation under task variation (Chua et al., 2023).

Our work takes an alternate route: instead of introducing a hierarchy of actions, DME builds hierarchy into the task representation. By learning a dynamic, multi-level latent belief, DME can embed a wide collection of task representations in a similar fashion, while keeping a single policy that sees structured uncertainty in task inference. In this sense, DME aims to capture many of the benefits of hierarchy through latent representation rather than through a system of policies.

## 4 DYNAMIC MIXTURE EMBEDDINGS

Contextual meta-reinforcement learning (meta-RL) requires task embeddings that rapidly adapt while distinguishing related tasks. Adaptive mixture priors offer a principled route to handling diverse and complex tasks that require more flexibility than what traditional fixed mixture priors can provide. However, implementing these priors in a manner bespoke for contextual meta-RL is not trivial. Not only does training need to be structured in a way such to reduce the risk of overfitting the high-capacity latent hierarchy, but similar latent representations must also induce comparable behaviour in the policy.

In this section, we introduce the key concepts underpinning our method DME. We begin by introducing the hierarchical Gaussian-mixture generative model at the core of DME, and discuss the training modules that integrate it into the contextual meta-RL process. From there, we describe the full DME algorithm and explain how virtual task simulation broadens the agent's experience during training. Finally, we briefly summarise key implementation details.

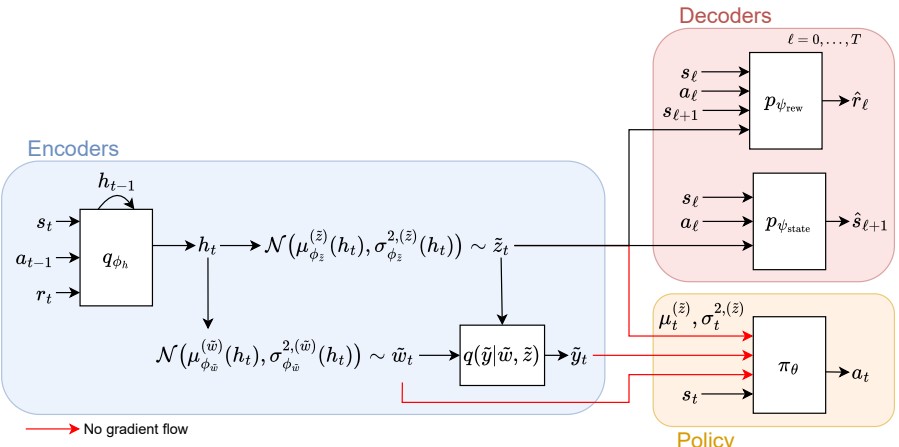

Figure 1: **DME overview.** Outline of a forward pass through DME and its hierarchical GMVAE encoder architecture. Online context is encoded into three latents: macro ($\tilde{w}$), mixture ($\tilde{y}$) and task ($\tilde{z}$). The RL policy utilises information from all three latents, including the task belief ($\mu^{(z)}, \sigma^{2,(z)}$). Only $\tilde{z}$ is used in the decoders, ensuring that the lowest-level latent contains all information necessary to reconstruct the MDP.

### 4.1 Hierarchical Gaussian Mixture Model for Dynamic Task Embeddings

**Latent hierarchy.** Inspired by the hierarchical variational model of Dilokthanakul et al. (2016), DME employs a three-tier latent structure $z = (\tilde{w}, \tilde{y}, \tilde{z})$. This consists of a continuous high-level macro-context latent $\tilde{w} \sim N(0, I)$ that dynamically controls mixture parameters, a discrete cluster latent $\tilde{y} \sim \text{Cat}(1/K)$ that partitions the task embedding space into semantically distinct regions, and a continuous low-level task embedding $\tilde{z}|\tilde{w}, \tilde{y} = k \sim N(m_\psi(\tilde{w}, k), v_\psi^2(\tilde{w}, k))$, where $k = 1, \ldots, K$, that ultimately infers the specific task.

A decoder-side network parameterised by $\psi$ maps the macro latent to mixture parameters, $\{(m_\psi(\tilde{w}, k), v_\psi^2(\tilde{w}, k))\}_{k=1}^K$, which define the component conditionals $p_\psi(\tilde{z}|\tilde{w}, \tilde{y} = k)$. The categorical $\tilde{y}$ selects an index $k$; given that index and $\tilde{w}$, $\tilde{z}$ is drawn from the corresponding Gaussian. In this sense, the macro latent $\tilde{w}$ parameterises the component Gaussians selected by $\tilde{y}$, determining in turn the means and variances of clusters in $\tilde{z}$-space. For the remainder of this section, we use $\phi$ for encoder parameters and $\psi$ for decoder parameters.

Critically, this hierarchical structure is explicitly dynamic. The macro latent $\tilde{w}$ actively repositions the Gaussian component means and variance structures in response to newly observed context. As a result, unlike the fixed cluster components learned by previous methods, mixture parameters naturally adapt and reallocate latent capacity in response to unseen task variation or distributional shift during training and testing.

**Task posteriors.** Unlike previous applications of this model - where training primarily seeks to fit observations drawn from a fixed generative prior - the objective of DME is to learn latent embeddings that quickly identify the underlying task given a small set of context transitions. Rather than shaping the prior to match observed data, the job of the variational model is to produce posteriors that capture sufficient task-relevant information to enable effective adaptation. Therefore, given the rolling context $\mathbf{c}_{0:t} = \{(s_\ell, a_\ell, r_\ell, s_{\ell+1})\}_{\ell=0}^t$, a recurrent encoder network parameterised by $\phi$ produces posterior distributions over the continuous latent variables:

$$q_\phi(\tilde{w}|\mathbf{c}_{0:t}) = N\big(\mu_\phi^{(w)}(\mathbf{c}_{0:t}), \sigma_\phi^{(w)2}(\mathbf{c}_{0:t})\big), \tag{1}$$

$$q_\phi(\tilde{z}|\mathbf{c}_{0:t}) = N\big(\mu_\phi^{(z)}(\mathbf{c}_{0:t}), \sigma_\phi^{(z)2}(\mathbf{c}_{0:t})\big). \tag{2}$$

Recurrent task encoders are utilised as they are better at exploiting temporal dependencies, which is a desirable property for on-policy meta-RL as it trains on sequential context.

Then, conditioning on inferred $(\tilde{w}, \tilde{z})$, the discrete posterior for $\tilde{y}$ can be computed analytically, avoiding high-variance gradient estimators:

$$q(\tilde{y} = k|\tilde{w}, \tilde{z}) = \frac{N\big(\tilde{z} \,\big|\, m_\psi(\tilde{w}, k), \, v_\psi^2(\tilde{w}, k)\big)}{\sum_{j=1}^{K} N\big(\tilde{z} \,\big|\, m_\psi(\tilde{w}, j), \, v_\psi^2(\tilde{w}, j)\big)}. \tag{3}$$

A low-variance estimator for the categorical cluster latent offers a crucial advantage over traditional neural estimators by keeping cluster assignments more stable during training, ensuring that downstream policy gradients are computed with minimal noise from label switching.

Parametrising the model this way reduces estimator variance and yields a single-path inference procedure: at test time, only the two Gaussian encoders $q_\phi(\tilde{w}|\mathbf{c}_{0:t})$ and $q_\phi(\tilde{z}|\mathbf{c}_{0:t})$ are required; the categorical $q(\tilde{y}|\tilde{w}, \tilde{z})$ is computed analytically, avoiding a separate network for $\tilde{y}$. Importantly, $q_\phi(\tilde{z}|\mathbf{c}_{0:t})$ is a single network, which mirrors the inference flow of unimodal methods such as VariBAD (Zintgraf et al., 2021a). However, DME's training through its hierarchical generative model encourages $\tilde{z}$ to take on a multimodal structure, retaining the benefits of mixture modelling without increasing inferential complexity at runtime.

At each timestep the encoder updates $q_\phi(\tilde{w}|\mathbf{c}_{0:t})$, $q_\phi(\tilde{z}|\mathbf{c}_{0:t})$, and the induced $q(\tilde{y}|\tilde{w}, \tilde{z})$ so the policy conditions on up-to-date beliefs rather than stale estimates.

**Task Reconstruction.** The low-level task embedding $\tilde{z}$ is trained via an MLP decoder to best reconstruct *all future rewards and transitions*, not just at the current timestep. This coerces $\tilde{z}$ to contain information that identifies the overall reward and transition function, allowing it to better reconstruct the true MDP. Concretely, sampling $\tilde{z}_t \sim q_\phi(\tilde{z}|\mathbf{c}_{0:t})$ from context-conditioned posterior, the decoder models the distribution over all future states and rewards

$$p_\psi\big(s_{t:T}, r_{t:T-1} \,\big|\, s_t, a_{t:T-1}, \tilde{z}_t\big) = \prod_{\ell=t}^{T-1} p_\psi\big(s_{\ell+1} \,\big|\, s_\ell, a_\ell, \tilde{z}_t\big) \, p_\psi\big(r_\ell \,\big|\, s_\ell, a_\ell, s_{\ell+1}, \tilde{z}_t\big), \tag{4}$$

where $T$ denotes the end of the current episode horizon. Decoding the full rollout forces $\tilde{z}$ to distil the information needed to recover the underlying task MDP dynamics and reward structure, rather than merely forecasting the next step, and provides the primary learning signal for the encoder–decoder.

Moreover, we choose to condition the decoder on $\tilde{z}$ alone, excluding the macro $\tilde{w}$ and cluster latent $\tilde{y}$. By omitting the higher-level variables from the reconstruction path we force the information bottleneck to reside in $\tilde{z}$, allowing $\tilde{w}$ and $\tilde{y}$ to play a predominantly organisational role and embed cluster information without entangling task dynamics across multiple latent representations.

**Variational objective.** With $\tilde{z}_t \sim q_\phi(\tilde{z}|\mathbf{c}_{0:t})$ and future indices $\ell \in \{t, \dots, T-1\}$, we introduce a prediction error for state and reward transitions as the primary training signal for our encoder:

$$\mathcal{L}_{\text{state}} = -\sum_{\ell=t}^{T-1} \log p_\psi\big(s_{\ell+1} \,\big|\, s_\ell, a_\ell, \tilde{z}_t\big), \tag{5}$$

$$\mathcal{L}_{\text{reward}} = -\sum_{\ell=t}^{T-1} \log p_\psi\big(r_\ell \,\big|\, s_\ell, a_\ell, s_{\ell+1}, \tilde{z}_t\big). \tag{6}$$

While predictive accuracy encourages precise information capture, we must also prevent latent embeddings from overfitting or completely collapsing into trivial forms. We assume independent priors $p(\tilde{w}) = N(0, I)$, $p(\tilde{y}) = \text{Cat}(1/K)$, and $p_\psi(\tilde{z}|\tilde{w}, \tilde{y}) = N\big(\mu_\psi(\tilde{w}, \tilde{y}), \sigma_\psi^2(\tilde{w}, \tilde{y})\big)$, and impose Kullback-Liebler (KL) constraints on each posterior:

$$\mathcal{L}_{\tilde{w}} = D_{\text{KL}}\big(q_\phi(\tilde{w}|\mathbf{c}) \,\|\, p(\tilde{w})\big), \tag{7}$$

$$\mathcal{L}_{\tilde{y}} = D_{\text{KL}}\big(q(\tilde{y}|\tilde{w}, \tilde{z}) \,\|\, p(\tilde{y})\big), \tag{8}$$

$$\mathcal{L}_{\tilde{z}} = D_{\text{KL}}\big(q_\phi(\tilde{z}|\mathbf{c}) \,\|\, p_\psi(\tilde{z}|\tilde{w}, \tilde{y})\big), \tag{9}$$

noting that the loss term for $\tilde{y}$ has a closed-form solution.

Combining the predictive reconstruction and KL terms, we define our full variational training objective (ELBO) as

$$\mathcal{L}_{\text{ELBO}} = \alpha_s \mathcal{L}_{\text{state}} + \alpha_r \mathcal{L}_{\text{reward}} + \beta_w \mathcal{L}_{\tilde{w}} + \beta_y \mathcal{L}_{\tilde{y}} + \beta_z \mathcal{L}_{\tilde{z}}, \tag{10}$$

where $(\alpha_s, \alpha_r, \beta_w, \beta_y, \beta_z)$ are fixed loss coefficients.

Intuitively, the reconstruction terms $(\mathcal{L}_{\text{state}}, \mathcal{L}_{\text{reward}})$ ensure task embeddings encode actionable information about present and future dynamics and rewards. The KL terms regularise the latent hierarchy: $\mathcal{L}_{\tilde{w}}$ centres the macro latent to minimise drift, $\mathcal{L}_{\tilde{y}}$ penalises deviation from a uniform categorical prior and encourages use of all clusters, while $\mathcal{L}_{\tilde{z}}$ aligns the encoder's posterior $q_\phi(\tilde{z}|\mathbf{c})$ with the mixture prior $p_\psi(\tilde{z}|\tilde{w}, \tilde{y})$.

## 4.2 DYNAMIC MIXTURE EMBEDDINGS FOR CONTEXTUAL META-RL

DME leverages the hierarchical GMVAE architecture introduced in the previous section to infer latent task representations encountered in contextual meta-RL settings. Once learned, the full latent $z = (\tilde{w}, \tilde{y}, \tilde{z})$ can serve as an informative summary of task information, guiding the agent's behaviour in unseen tasks and environments. Pseudocode for the full DME algorithm can be found in the Appendix.

**Policy conditioning.** Following prior work in on-policy contextual meta-RL, we feed the policy the entire $\mu_{\tilde{z}}, \sigma_{\tilde{z}}^2$ posterior over the task latent $\tilde{z}$, rather than sampling a single inferred embedding (Zintgraf et al., 2021b). In Bayesian literature this constitutes the agent's belief state, explicitly providing both the agent's current estimate of the task and the uncertainty surrounding the estimate. Supplying the belief state allows the agent policy to decide dynamically whether to explore and reduce task uncertainty, or to exploit the existing prediction.

However, relying solely on $\tilde{z}$ to influence agent behaviour provides little guidance about *where* in latent space would be most helpful to explore to reduce task uncertainty. Therefore, we also supply the policy with higher-level latents $\tilde{w}$ and $\tilde{y}$. At every environment step the encoder revisits the accumulating context and refreshes its posteriors over $\tilde{w}$, $\tilde{y}$, and $\tilde{z}$. Together, they provide structured cues that accelerate adaptation and empirically improve performance in difficult benchmarks, as was seen in Section 5.2.

**Virtual training.** To improve robustness and generalisation, DME follows the lead of Lee et al. (2023) and employs virtual training, periodically generating synthetic tasks using its learned hierarchical embedding and reward decoder. Virtual training seeks to address incomplete or insufficient task coverage by imagining plausible rewards for existing transitions under a different task embedding. These rewards are generated by sampling from the hierarchical prior:

$$\tilde{w}_{\text{virt}} \sim p(\tilde{w}),$$
$$\tilde{y}_{\text{virt}} \sim p(\tilde{y}),$$
$$\tilde{z}_{\text{virt}} \sim p_\psi(\tilde{z}|\tilde{w}_{\text{virt}}, \tilde{y}_{\text{virt}}).$$

Conditioned on the sampled virtual task representation $\tilde{z}_{\text{virt}}$, DME samples triplets $(s_\ell, a_\ell, s_{\ell+1})$ from the replay buffer and produces a synthetic reward

$$r_{\text{virt}} \sim p_\psi(r \mid s, a, s', \tilde{z}_{\text{virt}}). \tag{11}$$

In practice, these synthetic trajectories are sampled alongside real experience at some scheduling rate $\gamma \in [0, 1]$, which is increased over time.

## 4.3 IMPLEMENTATION DETAILS

DME employs PPO (Schulman et al., 2017) as the base RL algorithm. The on-policy PPO naturally suits scenarios involving continuously evolving task embeddings $\tilde{z}$, since updating directly from recent trajectories avoids the distributional mismatches that arise in off-policy contexts.

Given the DME latent decomposition $z = (\tilde{w}, \tilde{y}, \tilde{z})$, the loss functions for training PPO with dynamic mixture embeddings are given below

$$\rho_t = \frac{\pi_\theta(a_t|s_t, \tilde{w}_t, \tilde{y}_t, \tilde{z}_t)}{\pi_{\theta_{\text{old}}}(a_t|s_t, \tilde{w}_t, \tilde{y}_t, \tilde{z}_t)}, \tag{12}$$

$$\mathcal{L}_{\text{policy}} = -\mathbb{E}\big[\min\big(\rho_t A_t, \text{clip}(\rho_t, 1 - \epsilon, 1 + \epsilon)A_t\big)\big], \tag{13}$$

$$\mathcal{L}_{\text{value}} = \tfrac{1}{2}\mathbb{E}\big[(V_\theta(s_t, \tilde{w}_t, \tilde{y}_t, \tilde{z}_t) - \hat{V}_t)^2\big], \tag{14}$$

$$\mathcal{L}_{\text{entropy}} = -\beta_{\text{ent}}\mathbb{E}\big[\mathcal{H}(\pi_\theta(\cdot|s_t, \tilde{w}_t, \tilde{y}_t, \tilde{z}_t))\big]. \tag{15}$$

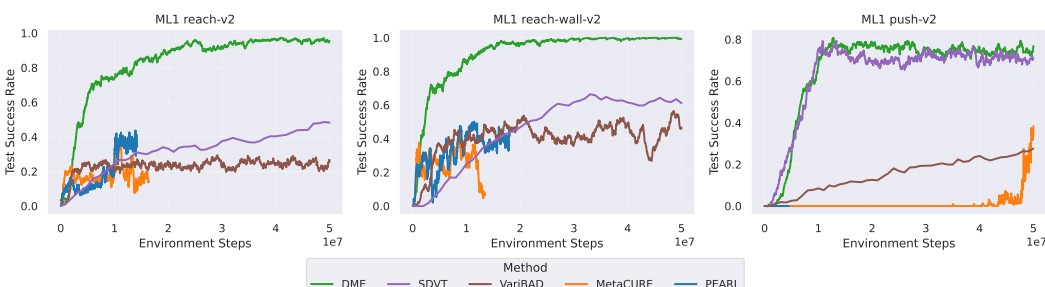

Figure 2: **MetaWorld ML1**. DME rapidly achieves near-perfect scores in the parametric single-environment MetaWorld ML1 benchmark, achieving a new state-of-the-art performance on those benchmarks.

where $\rho_t$ represents the policy likelihood ratio, $A_t$ is the advantage estimate and $\epsilon$ is the clipping constant.

The overall PPO loss,

$$\mathcal{L}_{\text{PPO}} = \mathcal{L}_{\text{policy}} + \mathcal{L}_{\text{value}} + \mathcal{L}_{\text{entropy}}, \tag{16}$$

is optimised separately from the variational objective, and we treat $(\tilde{w}, \tilde{y}, \tilde{z})$ as fixed conditioning variables (no gradient flow from $\mathcal{L}_{\text{PPO}}$ to $\phi$ or $\psi$).

## 5 EXPERIMENTS AND RESULTS

In this section, we describe experiments that aim to answer the following questions: 1) How does DME adapt to parametric and non-parametric variation? 2) Does including the macro latent $\tilde{w}$ in the policy input impact performance?

### 5.1 PARAMETRIC TASK ADAPTATION

**Experiment setup.** We begin by evaluating how well DME adapts to parametric task variation in order to understand whether DME's flexible GMVAE parameterisation can efficiently adapt to a smaller potential task space. The MetaWorld ML1 benchmark consists of a variety of separate environments where a robotic Sawyer arm must complete a certain task. For this experiment, we choose *reach-v2*, *reach-wall-v2*, and *push-v2* due to their frequency in related works. Each ML1 environment randomises a variety of parameters at the start of the episode. The agent must infer those parameters from a handful of transitions and subsequently solve the task.

We compare DME against four strong contextual meta-RL baselines that range across both on-policy belief-based inference (like DME), and off-policy sample-based inference:

- variBAD (Zintgraf et al., 2021a), which shares the same core belief-based inference framework as DME but with a unimodal Gaussian task embedding.
- SDVT (Lee et al., 2023), which also utilises virtual training, but with a fixed-prior GMVAE.
- PEARL (Rakelly et al., 2019), an off-policy method that, after gathering context with an uninformative prior, samples a single fixed posterior task embedding per adaptation episode.
- MetaCURE (Zhang et al., 2021), which extends the PEARL method with a separate explorer that maximises task information gained during initial exploration.

For belief-based agents (DME, variBAD, SDVT) we measure the return and task success rates of the final (third) episode, while for sample-based agents (PEARL, MetaCURE) we provide two exploratory episodes and measure the performance of the subsequent exploitation episode. We utilise the original source code and hyperparameters for all implementations.

**Results.** Final success rates are presented in Figure 2. DME establishes a new state-of-the-art across the suite, achieving near-perfect success rates on *reach-v2* and *reach-wall-v2* in particular. On *push-*

*v2*, it performs similarly to its closest peer SDVT. This is not surprising - SDVT utilises a similar Gaussian mixture prior, but its structure is fixed. We hypothesise that the increased representational capacity enables the macro latent $\tilde{w}$ to better learn the full range of parameterisation available, which allows the agent to completely solve the task.

Although the off-policy PEARL performs well in *reach-v2*, we note that wall-clock time for off-policy contextual meta-RL can be considerably longer due to the additional gradient steps taken per epoch. In addition, both PEARL and MetaCURE completely fail to solve *push-v2*, highlighting the fragility of their posterior sampling-based adaptation.

## 5.2 Non-Parametric Task Adaptation

**Experiment setup.** Although parametric adaptation remains a relevant challenge in complex environments, the true challenge for contextual meta-RL algorithms lies in adapting to *non-parametric* task variation and the qualitatively different goals and transition dynamics that may result.

In order to evaluate the performance of DME when exposed to non-parametric task variation, we utilise the challenging MetaWorld ML10 and ML45 benchmarks, whose training environment sets span 10 and 45 distinct manipulation tasks, respectively. Each task corresponds to a separate ML1 environment - ranging from standard motion (*reach-v2*), to manipulation tasks (*door-open-v2*) and more unusual objectives (*basketball-v2*). Beyond these structural and semantic differences, each task also randomises continuous parameters (object masses, friction coefficients, goal positions), offering a comprehensive test of an algorithm's capacity to adapt across both structural and parametric change.

After training, agents are evaluated on five held-out test tasks that were absent from the meta-training set but can, in principle, be solved by learning and applying the fundamental manipulation skills required to solve training tasks. We compare DME to the same peer baselines used in the parametric study — variBAD, SDVT, PEARL, and MetaCURE.

**Results.** Figure 3 reports success and return curves for both training and test tasks. Again, DME achieves the highest success rate on unseen test tasks, outperforming SDVT on both the ML10 and ML45 benchmarks. This improvement indicates that the ability to flexibly reallocate mixture means during training reduces over-specialisation and leaves extra capacity for genuinely novel behaviours.

This is further shown by how SDVT significantly outperforms DME in training environments, suggesting that the fixed mixture parameters of SDVT lead it to overfit on the training task distribution, leaving it unable to adapt to non-parametric task changes whose reward structure falls outside of its existing clusters. In contrast, the dynamic clusters of DME provide more capacity to generalise to unseen tasks at test time.

Only final results for the off-policy baselines, PEARL and MetaCURE, appear as they were tested for far fewer environment steps due to their off-policy nature. However, despite this, the wall-clock time remained comparable to the on-policy methods due to their tendency to take many more gradient steps per epoch. Despite this advantage in sample efficiency, both methods struggle to effectively adapt, possibly because their single-shot posterior sampling adaptation strategy is unreliable when needing to adapt to non-parametric changes in the task required.

## 5.3 Ablation Studies

**Policy input.** Although the macro latent $\tilde{w}$ is a crucial component of the hierarchical GMVAE, we seek to understand whether knowing $\tilde{w}$ is helpful for the RL agent policy $\pi_\theta$. We compare the full DME agent that passes $\tilde{w}$ to the policy, $\pi_\theta(a|s, \tilde{w}, \tilde{y}, \tilde{z})$, to a variant that does not do so, resulting in the policy $\pi_\theta(a|s, \tilde{y}, \tilde{z})$. We continue to condition the policy on $\tilde{y}$ as previous work has shown the benefits of including a discrete task- or subtask cluster as policy information (Lee et al., 2023).

In Figure 4, we see that conditioning on $\tilde{w}$ consistently improves performance relative to the $\tilde{w}$-ablated variant across both *ML10* and *ML45* benchmarks. The intuition is straightforward: while $\tilde{y}$ provides an explicit task cluster, the agent requires knowledge of the macro latent in order to understand where those clusters may be in embedding space. In this sense, both macro latent and cluster latent are required for the agent to truly understand mixture dynamics.

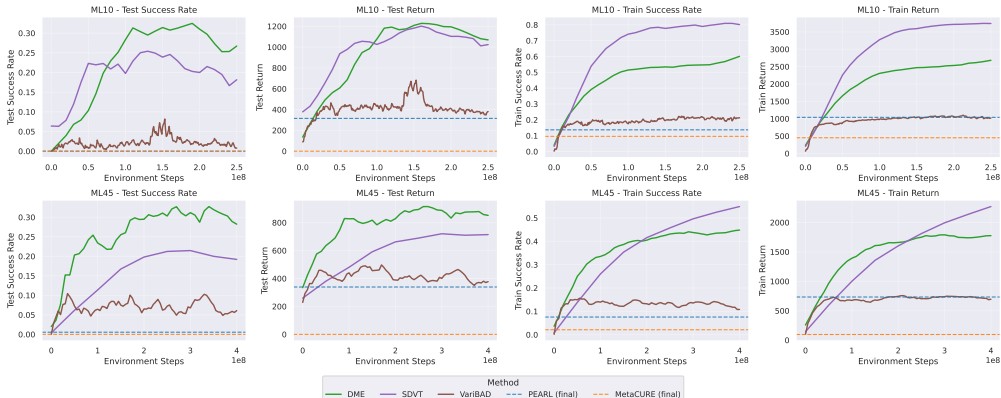

Figure 3: **Non-parametric MetaWorld**. DME outperforms contemporary contextual meta-RL methods on the difficult MetaWorld ML10 and ML45 benchmarks, achieving higher task success rates and returns. The lower success rates in training tasks suggest that DME's adaptive clusters are less likely to overfit on the training distribution.

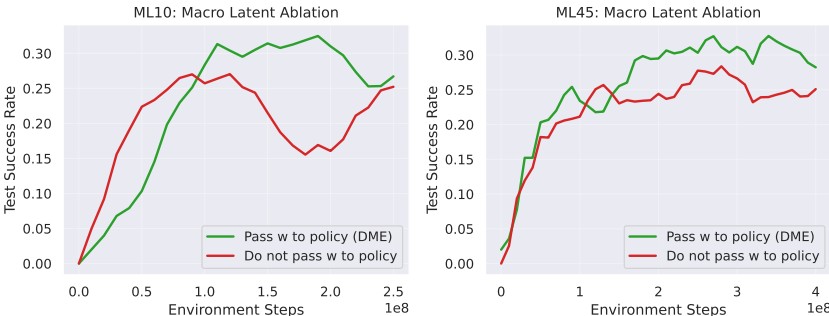

Figure 4: **Policy input ablation**. Passing the macro latent $\tilde{w}$ to the meta-RL policy $\pi_\theta(a|s, \tilde{w}, \tilde{y}, \tilde{z})$ improves performance, suggesting that $\tilde{w}$ contains information useful for adaptation.

## 6 CONCLUSION

In this paper we presented Dynamic Mixture Embeddings (DME), a contextual meta-RL algorithm that learns an adaptive Gaussian-mixture prior conditioned on a macro latent, so mixture components dynamically adapt as context is gathered. An analytic cluster assignment stabilises the hierarchy, while a decoder trained to predict entire future transitions and rewards pushes the continuous task code to capture MDP-level structure. Together, these choices deliver strong adaptation under both parametric and non-parametric variation, achieving state-of-the-art performance on the MetaWorld suite.

There are several promising directions for further study. First, there is still opportunity to investigate alternate formulations of the latent hierarchy, such as incorporating soft cluster assignments rather than relying solely on hard partitions. As this has proven successful in fixed-cluster context (Lee et al., 2023), we believe including the additional flexibility of DME-style may prove beneficial. In addition, there is certainly still room to refine the interaction between virtual training and the higher-level latents $\tilde{w}$ and $\tilde{y}$. Integrating virtual training into cluster allocation or enabling dynamic cluster counts for $\tilde{y}$ with a stick-breaking prior (Nalisnick & Smyth, 2017) could further improve the flexibility and robustness of this method.

Ultimately, this work has shown that dynamic hierarchical task representations are a practical and effective approach to improve adaptation and generalisation in contextual meta-RL. By utilising its adaptive mixture prior, DME demonstrates that representational flexibility matters for fast adaptation across both parametric and structural variation.

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

## A    APPENDIX: ALGORITHM

---

**Algorithm 1** DME

---

**Input:** Training task distribution $p(M)$, virtual rate $\gamma$
Initialise on-policy storage $B$; policy $\pi_\theta$; encoder $q_\phi$; decoder $p_\psi$
**while** not done **do**
    Sample a task $M \sim p(M)$
    *# On-policy data collection*
    Reset context $\mathbf{c} \leftarrow \{\}$
    Sample $z_0 = (\tilde{w}_0, \tilde{y}_0, \tilde{z}_0) \sim p(\tilde{z}|\tilde{y}, \tilde{w})p(\tilde{y})p(\tilde{w})$
    **for** timestep $t$ during rollout **do**
        Form policy belief $b_t = (\mu_{\tilde{z}_t}, \sigma_{\tilde{z}_t}^2, \tilde{w}_t, \tilde{y}_t)$
        Sample action $a_t \sim \pi_\theta(a_t|s_t, b_t)$ and step environment
        Append $(s_t, a_t, s_{t+1}, r_t)$ to $B$ and update $\mathbf{c}_{0:t}$
        Update $z_{t+1} = (\tilde{w}_{t+1}, \tilde{y}_{t+1}, \tilde{z}_{t+1}) \sim q_\phi(\tilde{w}|\mathbf{c}_{0:t})q_\phi(\tilde{z}|\mathbf{c}_{0:t})q(\tilde{y}|\tilde{w}, \tilde{z})$
    **end for**
    *# Training steps*
    Sample minibatches of sequential context $\mathbf{c} \sim B$ and an on-policy PPO batch $\mathbf{x} \sim B$
    Re-calculate posterior $z = (\tilde{w}, \tilde{y}, \tilde{z}) \sim q_\phi(\tilde{w}|\mathbf{c})q_\phi(\tilde{z}|\mathbf{c})q(\tilde{y}|\tilde{w}, \tilde{z})$
    Calculate $\mathcal{L}_{\text{state}}$ according to Equation 5
    **if** with probability $\gamma$ **then**
        *# Virtual update*
        Sample $(\tilde{w}_{\text{virt}}, \tilde{y}_{\text{virt}}, \tilde{z}_{\text{virt}}) \sim p(\tilde{z}|\tilde{y}_{\text{virt}}, \tilde{w}_{\text{virt}})p(\tilde{y})p(\tilde{w})$ from prior
        Calculate rewards $r_{\text{virt}}$ according to Equation 11
        Calculate $\mathcal{L}_{\text{reward}}$ using virtual rewards
    **else**
        Calculate $\mathcal{L}_{\text{reward}}$ according to Equation 5
    **end if**
    Calculate $\mathcal{L}_{\tilde{w}}$, $\mathcal{L}_{\tilde{y}}$, $\mathcal{L}_{\tilde{z}}$ according to Equations 7-9
    Calculate $\mathcal{L}_{\text{ELBO}}$ according to Equation 10
    Calculate $\mathcal{L}_{\text{PPO}}$ according to Equation 16
    *# Gradient update*
    Update $\psi, \phi$ by minimising $\mathcal{L}_{\text{ELBO}}$
    Update $\theta$ by minimising $\mathcal{L}_{\text{PPO}}$
    Empty $B$
**end while**

---

