# OpenReview forum: "Dynamic Mixture Embeddings for Contextual Meta-Reinforcement Learning"
_ICLR.cc/2026/Conference — Submitted to ICLR 2026_

### Official Review · Reviewer_BStz · 2025-10-21

**Soundness:** 2
**Presentation:** 2
**Contribution:** 2
**Rating:** 4
**Confidence:** 5

**Summary:**

This paper proposes a new task representation learning method for meta reinforcement learning (meta-RL), called Dynamic Mixture Embeddings (DME). The core idea is to dynamically represent task embeddings using a Gaussian mixture model structured by a three-tier latent hierarchy, denoted as $( \tilde{w}, \tilde{y}, \tilde{z} )$. Through this hierarchical representation, DME can capture multi-modal task distributions under both parametric and non-parametric task variations. Experimental results demonstrate that DME achieves strong performance on the MetaWorld benchmarks.

**Strengths:**

1. The authors clearly identify the limitation of unimodal task representations when dealing with non-parametric task structures.
2. They propose DME, a hierarchical task representation framework capable of modeling multimodal task distributions.
3. Extensive experiments are conducted to verify the effectiveness of the proposed method.

**Weaknesses:**

1. The paper lacks theoretical justification for the soundness and effectiveness of DME.
2. The explanation of the latent hierarchy and the corresponding teaser figure is confusing and not well aligned.
3. The reported experimental results for the most relevant baseline, SDVT, show a significant performance drop compared to its original paper.

**Questions:**

1. In Figure 1, the low-level latent variable $\tilde{z}$ appears to be directly produced by the encoder. I do not observe the hierarchical structure described in the first paragraph of Section 4.1.
2. While $\tilde{z}$ can indeed be updated via reconstruction loss and KL regularization, there is no theoretical analysis demonstrating that the learned $\tilde{z}$ possesses multimodal properties. For instance, given a trained encoder, $\tilde{z}$ seems to depend only on the input context, and not on the macro latent $\tilde{w}$ or the cluster latent $\tilde{y}$.
3. The role of the macro latent $\tilde{w}$ is unclear. Could it be replaced by the hidden state $h_t$? Is $\tilde{w}$ truly necessary?
4. In the MetaWorld ML10 and ML45 experiments, the SDVT baseline shows a large performance drop during testing, despite having similar training performance to its original paper. This discrepancy should be analyzed and explained.

---

> ### Author Response · Authors · 2025-12-01
>
> Firstly, we thank Reviewer BStz for the detailed review.
>
> Based on all reviews received, we have decided to take some additional time to work on improving DME and the paper further before seeking publication, with a particular focus on improving the breadth of experimental comparisons of our method. We will also investigate additional theoretical analysis.

---

### Official Review · Reviewer_3i9F · 2025-10-29

**Soundness:** 2
**Presentation:** 1
**Contribution:** 2
**Rating:** 2
**Confidence:** 5

**Summary:**

The paper Dynamic Mixture Embeddings for Contextual Meta-Reinforcement Learning proposes a new method, DME, to improve task adaptation in meta-reinforcement learning. Traditional approaches often use a single, fixed prior that cannot capture complex or multi-modal task structures. DME introduces a hierarchical Gaussian-mixture variational autoencoder, where mixture components are dynamically adjusted based on a high-level latent variable. This structure allows the task representation to change and grow as new context is gathered, enabling better handling of unseen or non-parametric variations. DME also incorporates virtual training, where synthetic tasks are generated from the learned latent model to improve generalization. Experiments on the MetaWorld benchmark show that DME achieves state-of-the-art performance across both simple parametric and complex non-parametric task variations, outperforming previous methods such as VariBAD, SDVT, and PEARL in author given tasks.

**Strengths:**

The main strength of this paper lies in its novel hierarchical latent structure that dynamically adapts to new task information, providing greater representational flexibility than previous fixed-mixture approaches. DME successfully unifies belief-based task inference, adaptive latent modelling, and virtual training into one coherent framework. This combination enables the agent to maintain stable learning while effectively capturing diverse task variations. The experiments  conduct in Meta-world robot manipulation that covers both parametric and non-parametric tasks and including ablation studies that clarify the role of latent variable. The method’s ability to outperform recent baselines across MetaWorld settings demonstrates its robustness. Overall, DME represents an advancement in contextual meta-reinforcement learning by showing how adaptive latent mixtures can improve generalization and fast adaptation in robot manipulation tasks.

**Weaknesses:**

1.The author use a hiearchical Gaussian mixture VAE as task inference model. However, such prior model is still within Bayesian parametric domain, which means that they must predefine the number components and also other parameters within its latent space [1], Such configuration will face significant defect when handling non-stationary scenarios or continus stream of tasks. Such scenarios is also interest for meta-RL. Moreover, using Gaussian mixture (GM) for latent task inference at upstream is nothing new, previous work such as CEMRL [2] has already employ GM as prior for meta-RL, this reduce the overall contribution of this paper. Furthermore, based on this, recent advances have explored using Bayesian non-parametric model as prior, examples like MELTS [3] employ Dirichlet process mixture model to describe the task context, where the number, position, shape, and density of each component can be dynamically determined online without predefinition or throughlt finetuning. Overall, this paper offers limited innovation and can only be considered an incremental contribution.

2.The experimental results are unreliable. First, the author claimed that their DME can dynamically adapt shape and position according to the context, however they didn't provide any qualitative or quantitative results to support their claim (e.g., t-sne plot), so it is unclear whether such claim ceratinly same as author argued. Second, all experimental results were run only once, making it impossible to determine the extent of any bias in the findings or whether the results were cheery-picked by the authors. Third, Some experiment details are missing, making it diffcult to judge whether the setup is fair. For example, in the parametric task, how the author defined their training task distribution and test set distrubution? In the non-parametric tasks, which tasks are selected as train set, and which are selected as test set? Fourth, Some key ablation study are missing. The author claimed that their framework can rely on small slice of transition for context learning, but how small is so-called small? And author used LSTM for recurrent network, what about GRU network, which can perform better? There are too many uncertain points remain for answer.

3.The writing is obscure and difficult to understand, for example: 1) The article frequently uses long, complex sentences. 2) Some transition words are used in an unusual manner. I understand the author may have used an LLM to polish the article, but please have a native English speaker review and refine it after the polishing. 3) Some key information and background are missing (e.g., the derivation of the hierarchy GMVAE), which makes some terms used in the paper are difficult to understand.

In summary, given the limited innovation of this paper, the absence of its key experiments, and the overall inadequacy of experimental details, I can only conclude that this is an UNFINISHED work. Therefore, I must issue a rejection decision.

Reference:

[1] Nat Dilokthanakul, Pedro AM Mediano, Marta Garnelo, Matthew CH Lee, Hugh Salimbeni, Kai
Arulkumaran, and Murray Shanahan. Deep unsupervised clustering with gaussian mixture variational autoencoders. arXiv preprint arXiv:1611.02648, 2016.

[2] Zhenshan Bing, David Lerch, Kai Huang, and Alois Knoll. Meta-reinforcement learning in nonstationary and dynamic environments. IEEE Transactions on Pattern Analysis and Machine Intelligence, 45(3):3476–3491, 2023.

[3] Zhenshan Bing, Yuqi Yun, Kai Huang, and Alois Knoll. Context-based meta-reinforcement learning with bayesian nonparametric models. IEEE Transactions on Pattern Analysis and Machine Intelligence, 46(10):6948–6965, 2024.

**Questions:**

1.Some key backgrounds are missing. For example, lack of introduction about Hirarchical Gaussian-mixture VAE in preliminaries or appendix.

2.The experiments runs only once, author needs to run at least with 5 random seeds, reporting its mean and std as evalution metrics.

3.Explicit visualization of latent space for demonstraing context adaptation is necessary. can author provide for example t-sne or pca plot of latent space?

4.Some key ablation studies are missing. For example, to what extend that context length may influent the task inference? Is LSTM network the optimal choise for context inference?

5.For parametric tasks evaluation, select only three tasks for evaluation is insufficient. I think at least 10 distinct tasks and embodied settings are necessary for generalization and scalability validation (e.g., RLBench, Mujoco locomotion tasks). Again, for non-parametric tasks, author also needs to conduct more experiment on more benchmarks (RLbench or Mujoco) to prevent overfitting on the given meta-world benchmarks.

6.How are the train set and test set in the the experiment set up? author needs to clarify this. The authors should clarify how the training and test sets are constructed and configured in the experimental setup.

7.The authors needs to compare with state-of-the-art context meta-RL frameworks, including CEMRL that also employ Gaussian mixtures from Bayesian parametric domain, and MELTS that use DPMM from Beyasian non-parametric domain as prior for contextual inference.

---

> ### Author Response · Authors · 2025-12-01
>
> Firstly, we thank Reviewer 3i9F for the detailed review.
>
> Based on all reviews received, we have decided to take some additional time to work on improving DME and the paper further before seeking publication, with a particular focus on improving the breadth of experimental comparisons of our method, including additional benchmarks outside of the MetaWorld suite. We appreciate the advice in terms of better illustrating behaviour through visualisation of the latent space.

---

### Official Review · Reviewer_mbvV · 2025-10-30

**Soundness:** 2
**Presentation:** 2
**Contribution:** 1
**Rating:** 2
**Confidence:** 4

**Summary:**

This paper proposes Dynamic Mixture Embeddings (DME), a belief-based contextual meta-RL method that embeds task information with a hierarchical GMVAE: a macro latent $\tilde{w}$ modulates Gaussian-mixture component means/variances, a discrete cluster latent $\tilde{y}$ indexes components, and a continuous task latent $\tilde{z}$ captures MDP-level structure. The key technical structure is to condition mixture parameters on $\tilde{w}$, allowing component locations and scales to adapt online as context accumulates; the categorical posterior $q(\tilde{y}=k|\tilde{y},\tilde{z})$ is computed analytically for low-variance calculation. The decoder is trained to reconstruct full future trajectories and rewards conditioned only on $\tilde{z}$, enforcing an information bottleneck that pushes task-relevant dynamics into the continuous code. The method also incorporates virtual training by sampling synthetic task latents from the adaptive prior. Empirically, DME claims state-of-the-art ML1 performance and improved generalization on ML10/ML45 test tasks in MetaWorld, with an ablation indicating the usefulness of passing $\tilde{w}$ to the policy. Overall, the paper’s depth lies in the hierarchical latent design, the analytic cluster assignment, and its integration with belief-based conditioning and virtual training.

**Strengths:**

* **S1.** This paper addresses an important field in Meta-RL. MetaWorld ML10 and ML45 are Non-parametric and Out-Of-Distribution task environments, and they remain domains that have not yet been fully solved. Because these settings consider the distribution of tasks that real robotic agents would perform, I believe they should be treated as important topics in Meta-RL; however, they are relatively difficult to tackle and therefore seem to have been studied less frequently. Even under these circumstances, the authors address the Non-parametric task problem and, by releasing the related code, I believe they are making efforts for future researchers.

* **S2.** The overall architectural design proposed by the authors does not appear to be difficult. The authors simply modified the structure of the existing GMVAE to enable the parameters to vary dynamically. As architectures in the Meta-RL field have been becoming increasingly complex recently, I believe the authors have attempted to address this with a simple design.

**Weaknesses:**

* **W1.** I respectfully think that the claimed novelty and the validation thereof are not particularly strong. The proposed method parameterizes the prior of GMAVE so that it can be adaptively adjusted; however, this architecture and following equations seem to be directly borrowed from the existing structure[1] that the authors cite at line 187. Even if, as noted at line 115, this structure is applied to meta-RL for the first time, the structure itself already exists in prior work, and I think there does not appear to be additional architectural component introduced specifically to solve the MetaWorld benchmarks. To my knowledge, the Task Reconstruction method at line 234 is the technique used in VariBAD[2], and the belief method at line 284 as well as the virtual training method at line 296 also follow prior works[2, 3] as the authors state. Moreover, the validation related to these claims seems insufficient. For example, at lines 69 and 412 the authors assert that, owing to this architecture, “dynamic mixture representation achieves greater expressive capacity,” yet there seems to be inadequate evidence that this is actually the case. Most Meta-RL studies -including SDVT[3], which appears to be the authors’ baseline- visualize the learned representations to substantiate such claims. By contrast, the authors present only performance metrics, which might weaken their argument.

* **W2.** The problem formulation raised by the authors appears unclear. At lines 58 and 201, they claim that prior work suffers from the drawback that the prior over mixture parameters is fixed, but they do not appear to verify that this is indeed problematic. Providing theoretical justification or empirical evidence would clarify and strengthen the issue they raise.

* **W3.** I have concerns regarding the performance and effectiveness of DME. I agree with the authors that improving test performance is important in Meta-RL. However, I do not believe that increasing test performance while sacrificing training performance is particularly meaningful in Meta-RL. In my view, the essence of Meta-RL is to be able to solve the training tasks while also solving the test tasks. Furthermore, relative to recent online Meta-RL works on the MetaWorld benchmarks, the training performance appears inferior. AMAGO2[4] and ECET[5] achieve training performance of approximately ≥ 0.9, and in particular ECET achieves test performance averaging ≥ 0.3. In comparison. DME seems to have lower training performance than the state of the art algorithms, suggesting that the proposed hierarchical latent structure may not be effective for the training in MetaWorld benchmarks.

* **W4.** The explanations of terminology do not appear sufficiently detailed. For example, the authors refer to $\tilde{w}$ as a “macro latent,” but there is no clear definition of what “macro latent” means or some examples of that. They also refer to $\tilde{y}$ as a “discrete cluster latent,” yet they do not explain what exactly is being represented as clusters. I think this lack of clarity can render the authors’ claims ambiguous.

* **W5.** I have concerns regarding experimental reproducibility. First, there is no description of the hyperparameters used by the authors. For instance, the number of clusters appears to be denoted by $K$, but there is no explanation of what entities are being clustered or what value the authors set for $K$. The authors do not provide criteria for setting the loss coefficients($\alpha_s, \alpha_r, \beta_w, \beta_y, \beta_z$) in Equations (10) and (15). As the representations may be likely sensitive to the latent dimensionality, the chosen latent dimensions and the rationale for those choices should be specified. The architectures of the GMVAE, decoder, and policy networks, as well as the optimizer and learning rate settings, are not reported. Second, I am concerned about the reliability of the reported performance curves. In Figures (2), (3), and (4), the authors do not state whether the experiments were conducted with a single seed or whether the results are averages over multiple seeds. Moreover, the authors does not provide the performance of all individual tasks in ML10 and ML45 environments, which might also weaken the reliability of the main performance graph(Figure (3)). Although the authors have released their code, it would be helpful if the criteria for hyperparameter settings and the experimental setups were reported in the main text or in the appendix.

[1] Dilokthanakul, Nat, et al. "Deep unsupervised clustering with gaussian mixture variational autoencoders." arXiv preprint arXiv:1611.02648 (2016).

[2]  Zintgraf, Luisa, et al. "VariBAD: A Very Good Method for Bayes-Adaptive Deep RL via Meta-Learning." International Conference on Learning Representations.

[3] Lee, et al. "Parameterizing non-parametric meta-reinforcement learning tasks via subtask decomposition." Advances in Neural Information Processing Systems.

[4] Grigsby, Jake, et al. "Amago-2: Breaking the multi-task barrier in meta-reinforcement learning with transformers." Advances in Neural Information Processing Systems.

[5] Shala, Gresa, et al. "Efficient Cross-Episode Meta-RL."  International Conference on Learning Representations.

**Questions:**

* **Q1.** I would appreciate if you could provide responses to every items described in the Weaknesses section.

* **Q2.** At line 383, the authors state that “we note that wall-clock time for off policy contextual meta-RL can be considerably longer due to the additional gradient steps taken per epoch,” but I am unable to find any report substantiating this. Is there a wall-clock time comparison experiment across algorithms?

---

> ### Author Response · Authors · 2025-12-01
>
> Firstly, we thank Reviewer mbvV for the detailed review.
>
> Based on all reviews received, we have decided to take some additional time to work on improving DME and the paper further before seeking publication, with a particular focus on improving the breadth of experimental comparisons of our method, including against the new AMAGO and ECET baselines. In addition, later versions of our work will seek thttps://www.nobodysausage.com/o provide more detailed illustration around the issues we seek to address.

---

### Official Review · Reviewer_et8R · 2025-10-30

**Soundness:** 3
**Presentation:** 3
**Contribution:** 2
**Rating:** 6
**Confidence:** 4

**Summary:**

This paper introduces Dynamic Mixture Embeddings (DME), a novel method for contextual meta-reinforcement learning that addresses the challenge of adapting to both parametric and, more importantly, non-parametric task variations. The core issue identified is that existing methods using Gaussian Mixture priors for task embeddings suffer from rigidity, as their mixture components (means, variances) are fixed after training. DME overcomes this by learning a hierarchical, adaptive mixture prior. Experimental results demonstrate that DME achieves SOTA performance across the challenging MetaWorld benchmark suite.

**Strengths:**

Strength
1. This paper is well-written and easy to read.
2.  The introduction of a dynamic, context-conditioned mixture prior is a great contriibution since it goes  beyond fixed clusters.
3. This paper conducted various experiments to verify the effectiveness of its method.

**Weaknesses:**

1. Insufficient Ablation on Virtual Training. It is unclear how much of DME's performance is driven by the novel dynamic prior versus the well-established benefits of virtual training.
2. Important Omission: The works by Bing et al. [1][2] are highly relevant. Specifically, "Context-Based Meta-Reinforcement Learning With Bayesian Nonparametric Models" (Bing et al., 2024) presents a compelling alternative by using a Bayesian nonparametric (BNP) model to automatically infer the number of mixture components KK from data, directly tackling the "rigidity" of fixed-KK GMMs. The absence of a comparison with such methods is a notable gap.
3. The experimental results, as presented in the learning curves (e.g., for ML1 tasks), appear to be based on a single training run per method. This is a limitation given the well-documented high variance inherent in deep reinforcement learning and meta-RL training.

[1] Bing, Zhenshan et al. “Context-Based Meta-Reinforcement Learning With Bayesian Nonparametric Models.” IEEE Transactions on Pattern Analysis and Machine Intelligence 46 (2024): 6948-6965.

[2] BZ. Bing, D. Lerch, K. Huang, and A. Knoll, “Meta-reinforcement learning in non-stationary and dynamic environments,” IEEE Trans. Pattern Anal. Mach. Intell., vol. 45, no. 3, pp. 3476–3491, Mar. 2023

**Questions:**

1. The results show that DME significantly outperforms VariBAD on parametric ML1 tasks (e.g., reach-v2), which are often considered unimodal. Could the authors explain why a dynamic mixture model provides such a strong advantage in a setting where a unimodal prior (like VariBAD's) should be sufficient?
2. The empirical results on the parametric ML1 tasks indicate that on-policy methods (DME, VariBAD) achieve higher success rates and more data efficient compared to off-policy methods like PEARL and MetaCURE. This is somewhat counter-intuitive, as off-policy algorithms are typically designed for greater sample efficiency by reusing past data. Could you provide your analysis on why this might be the case in the Meta-World benchmark?
3. The three-level latent structure (w, y, z) is a core contribution. Beyond the performance improvement shown in the ablation, is there any intuitive or qualitative interpretation of what the macro latent w and the cluster latent y are capturing
4. The core innovation of DME is its dynamic, multimodal task representation. To better understand how this structure facilitates adaptation, could you provide a qualitative visualization of the latent space?

---

> ### Author Response · Authors · 2025-12-01
>
> Firstly, we thank Reviewer et8R for the detailed review.
>
> Based on all reviews received, we have decided to take some additional time to work on improving DME and the paper further before seeking publication, with a particular focus on improving the breadth of experimental comparisons of our method. We will definitely include ablations on key structural decisions (e.g. Virtual Training), as well as further illustrative experiments to clarify the importance and intuition of the three-level latent structure.

---

### Meta-Review · Area_Chair_BxrT · 2025-12-07

**Summary:**

**Paper Summary**

This paper proposes DME, a new contextual meta-RL method consisting of a heirarchical Gaussian-mixture VAE and the virtual traininig strategy. This method aims to address the issue that existing methodologies rely on predefined or fixed priors when recognizing the current task. Experiments on MetaWorld suggest improved performance of DME compared to four other methods.

---

After reading the paper and reviewers' comments, the AC summarizes the paper's strengths and weaknesses below.

**Strengths**
- The paper studies a practical and interesting problem. Out-of-distribution scenarios, not only parameter changes, are easy to encounter during real-world deployment of learning based policies.
- A new contextual meta-RL method is introduced, aiming to address fixed or pre-known priors.

**Weaknesses**
- The paper clarity should be further improved. Even though some reviewers mentioned that the paper is well written, after reading the paper myself, I found several paragraphs that are not clear enough, particularly Section 4. Besides, there are several typos scattered in the draft, which should be corrected.
- The evaluation is not sufficient. The ablation on whether passing the macro is too naive. There are many other aspects worth exploring, such as why we should assume that the prior is formed by a Gaussian mixture model. The compared baselines are a bit old. The experimental results are provided without statistical information. The budget to interact with the environments is not equal for different methods, which raises concerns about unfair comparison.

**Reviewer Concerns:**

As the authors indicated that they decided to strengthen and polish the work following the reviewers' suggestions before seeking publication, and since they did not provide a rebuttal for this version of the submission, the AC therefore believes that all concerns raised by the reviewers remain unresolved.

**Reviewer Scores:**

The paper initially received scores of [2, 2, 4, 6]. As mentioned above, all concerns still remain unresolved because the authors did not provide a rebuttal, and the AC believes it is unlikely that the paper’s scores would increase even with a full author and reviewer discussion.

---

### Decision · Program_Chairs · 2026-01-26

Reject